# Variation in local trust Do Not Attempt Cardiopulmonary Resuscitation (DNACPR) policies: a review of 48 English healthcare trusts

Karoline Freeman,[1] Richard A Field,[1,2] Gavin D Perkins[1,2]

▶ Prepublication history and additional material is available. To view please visit the journal (http://dx.doi.org/10.1136/bmjopen-2014-006517).

[1]Division of Health Sciences, Warwick Medical School, University of Warwick, Coventry, UK
[2]Heart of England NHS Foundation Trust, Bordesley Green East, Birmingham, UK

**Correspondence to**
Dr Gavin D Perkins;
g.d.perkins@warwick.ac.uk

## ABSTRACT

**Objectives:** To explore Do Not Attempt Cardiopulmonary Resuscitation (DNACPR) policies from English acute, community and ambulance service Trusts for evidence of consistency and variation in implementation of national guidelines between healthcare organisations.

**Setting:** Acute, community or ambulance National Health Service (NHS) Trusts in England.

**Participants:** 48 NHS Trusts.

**Interventions:** Freedom of information requests for adult DNACPR policies were sent to a random sample of Trusts.

**Outcomes:** DNACPR policies were assessed on aspects identified from national guidelines including documentation, ethical and legal issues, decision-makers and involvement of others in DNACPR decisions as well as practical considerations such as validity, review and portability of decisions.

**Results:** Policies from 26 acute, 12 community and 10 ambulance service Trusts were reviewed. There was variation in terminology used (85% described documents as policies, 6% procedures and 8% guidelines). Only one quarter of Trusts used the recommended Resuscitation Council (UK) record form (or a modification of the form). There was variation in the terminology used which included DNAR, DNACPR, Not for CPR and AND (allow natural death). Accountability for DNACPR decisions rested with consultants at all acute Trusts and the most senior clinician at community Trusts. Most Trusts (74%) recommended discussion of decisions with a multidisciplinary team. Compliance with guidance requiring clinical staff to assess the patient for capacity and when to consult a lasting power of attorney or independent mental capacity advocate occurred less commonly. There was wide variation in the duration of time over which a DNACPR decision was considered valid as well as in the Trusts' approach to reviewing DNACPR decisions. The level of portability of DNACPR decisions between healthcare organisations was one of the greatest sources of variation.

**Conclusions:** There is significant variation in the translation of the national DNACPR guidelines into English healthcare Trusts' DNACPR policies.

## BACKGROUND

Decisions relating to Do Not Attempt Cardiopulmonary Resuscitation (DNACPR)

### Strengths and limitations of this study

- Key areas for strengthening current approaches were identified and included improving consistency for recording decisions and ensuring decisions are transferable between healthcare settings.
- Whether improving consistency in Do Not Attempt Cardiopulmonary Resuscitation policies will translate to improved implementation into practice and patient experience requires further study.
- The findings require confirmation in other settings.

are complex and require tried and tested decision-making processes as well as handover systems involving the patient, relevant others and a multidisciplinary healthcare team. Initiation of discussions of DNACPR is likely to occur if there is clinical evidence that CPR would be futile, that resulting harm would outweigh potential benefits or if a patient refuses CPR treatment. DNACPR decisions do not involve decisions about any other acute, life-saving treatments. In England the Human Rights Act 1998[1] and the Mental Capacity Act 2005[2] provide the legal basis for DNACPR decision-making. The Human Rights Act 1998[1] covers fundamental rights such as the right to life, the right to be free from inhuman and degrading treatment and the right to hold opinions and to receive information. The Mental Capacity Act 2005[2] works under the assumption that every adult is able to make their own decisions unless a mental capacity assessment shows otherwise. The latter sets out how to carry out an assessment of capacity, describes who can make decisions for people who lack capacity and provides a checklist to ensure that any decision taken on behalf of a person without capacity is in their best interest. On the basis of these laws, national guidelines[3] in England describe the context, setting and process for making

informed decisions to omit CPR and provide a framework to support decisions relating to CPR. Nonetheless, recent high profile cases in the media suggest that issues exist in terms of DNACPR decision-making in English healthcare settings claiming lack of consistency in approach across England. It is, therefore, of interest to understand the impact that the national guidelines have on local policies which, in the absence of a national DNACPR policy, determine local practice. By taking the view that current guidelines are informed primarily by ethical and legal considerations with little focus on the available research evidence there might be room for improvement in the practical guidance available. Furthermore, the guidelines provide general principles that require tailoring to local circumstances, which suggests that there may be room for interpretation of national guidelines when implemented into local policy. We therefore reviewed a random sample of local DNACPR policies from acute, community and ambulance services Trusts across England and mapped them against aspects from national guidelines to identify variation and consistencies between and within Trust types and inconsistencies in implementation of national guidelines.

## METHODS

Using the National Health Service (NHS) service directory,[4] we obtained NHS care Trust lists of acute, community and ambulance service (AS) services. We identified a random sample of 20 acute hospital Trusts for review: Trust lists were numbered and, using a random number generator in EXCEL (using the RAND and INDEX functions), a random list without duplication was created. The first 20 trusts on this list were included in the review. After mapping a further six Trusts were chosen to ensure geographical coverage. All 10 AS Trusts and a random sample of 12 community Trusts were further included using the same method as described for acute Trusts. Freedom of information requests for adult DNACPR policies were sent to the sample of English acute Trusts, ambulance service Trusts and community Trusts. If a separate DNACPR policy was not available the Trust's resuscitation policy was requested. Follow-up emails were sent once if (1) the wrong policy was sent, (2) additional information was apparently available on reading of the policy or (3) the health Trust had not responded after 2 months.

Additional information given in the accompanying emails by the Trust on further policies and validity of the policy was considered in the data extraction process as was information on the back of the DNACPR form if provided.

A data extraction form was established using aspects from the joint statement by the Resuscitation Council (RC (UK)), the British Medical Association (BMA) and the Royal College of Nursing (RCN) on decisions relating to cardiopulmonary resuscitation.[3] Aspects were chosen that were deemed important for DNACPR decision-

making by the expert advisory group including national leads in DNACPR guidelines, practicing clinicians in acute, community and palliative care as well as patient representatives. The resulting extraction form was piloted by two reviewers and was subsequently adjusted. The final form included 26 questions on aspects taken from the national guidelines including documentation, ethical and legal issues, DNACPR decision-makers and involvement of others in DNACPR decisions as well as practical considerations such as validity, review period and portability of decisions (see electronic supplement for data extraction table E1). Policies were then read in their entirety by one researcher (KF) and data extracted using the agreed form. Queries during the data extraction process were discussed and agreed with GDP. Thirty per cent of data extractions were checked by a second reviewer (RF) and confirmed as accurate, therefore further checking was not undertaken.

Data were synthesised quantitatively. Quantitative synthesis involved the reporting of frequencies of responses per question in tables for the paragraphs on documentation of DNCAPR decisions, implementation of legal and ethical aspects, decision-makers and involvement of others. These covered the questions that involved answers chosen from a list of limited options (eg, yes/ no/uncertain). For the questions concerning the validity, review and portability of decisions the review of policies identified substantial variation in local policy documents. Comprehensive data extraction was undertaken to capture all identified variation in the policies in these three areas in separate documents using descriptive coding. The main categories of the topic areas were then identified by discussion through clustering of the codes between the three authors. The categories were presented to the project and advisory groups for consensus and subsequently formed the basis for possible responses to the questions on the data extraction form. In order to retain the breadth and depth of the variation within the categories the data presented in the paragraph 'practical considerations' was synthesised narratively. The major areas of interest included the validity, review and portability including handover of decisions.

## RESULTS

A total of 48 local DNACPR policies were reviewed (100% response rate). The 26 acute Trusts consisted of six teaching hospitals, 19 district general hospitals and one specialist centre. Good geographical coverage of England was achieved with the reviewed policies from 26/156 acute and 12/24 community Trusts (figure 1). All 10 English AS were considered.

### Documentation

The Trusts produced specific local documents referred to as policies (85%), procedures (6%) or guidelines (8%) (table 1). There was variation in the terms used to describe resuscitation decisions. While the national

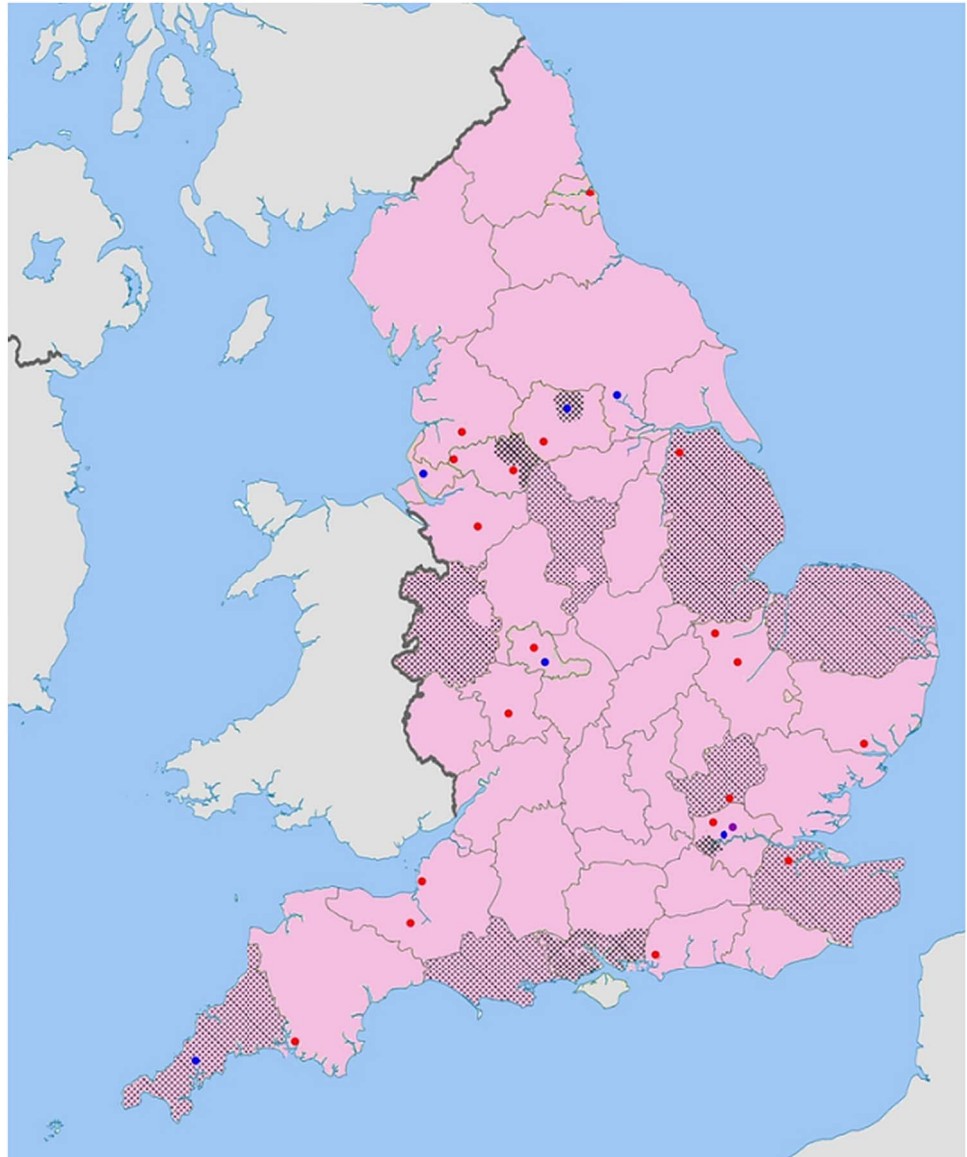

**Figure 1** Overview and geographic coverage of local Trust DNACPR policies included in the review. (● Teaching hospital; ● District general hospital; ● Specialist hospital; ▨ Community healthcare Trust).

guidelines recommend the use of the phrase 'do not attempt cardiopulmonary resuscitation' to avoid confusion,[3] two-thirds of Trusts referred to the term 'DNACPR', a quarter to 'DNAR' and the remainders to 'Not for CPR' or 'Allow Natural Death'. Half the Trusts had reviewed the relevant policy/guideline/procedure (referred to from here under the uniform term 'policy') within the preceding 12 months (range 1–47 months). One quarter of policies were outside the review period set by the Trust while half indicated the expired policy was currently under active review. Three quarters of policies reported having undergone an equality and diversity assessment.

There was variation in how DNACPR decisions were recorded (table 2). The guidelines clearly state that there is some benefit in using a standardised form and recommend the adoption of the RC (UK) model form.[3] In contrast, only one quarter of Trusts used the RC

(UK) DNACPR form or a modified version. As a general guide, modifications included adding or changing a couple of questions but required the general layout to be the same as the RC (UK) form while any more substantial changes would make the form bespoke. The majority of forms were paper forms (81%) with only 8% of Trusts using electronic versions. Healthcare professionals making the DNACPR decision were obligated to sign the form, which also had room for a countersignature by the consultant if a junior doctor initiated the decision which needs endorsement. In one specific case the form required that the doctor's signature must be witnessed by the registered nurse who was involved in the discussion on the DNACPR decision.

### Ethical and legal basis for DNACPR decisions

The joint statement 'Decisions relating to Cardiopulmonary Resuscitation' from the British Medical

**Table 1**  Overview of variation in type of local DNACPR policies by Trust type

| Policy type item | Acute Trusts (n=26), n (%) | Community Trusts (n=12), n (%) | AS Trusts (n=10), n (%) | Total, n=48, n (%) |
|---|---|---|---|---|
| Type of document | | | | |
| Policy | 22 (85) | 12 (100) | 7 (70) | 41 (85) |
| Procedure | 2 (8) | 0 | 1 (10) | 3 (6) |
| Guideline | 2 (8) | 0 | 2 (20) | 4 (8) |
| Terminology | | | | |
| DNAR | 6 (23) | 3 (25) | 3 (30) | 12 (25) |
| DNACPR | 16 (62) | 8 (67) | 7 (70) | 31 (65) |
| Not for CPR | 3 (12) | 0 | 0 | 3 (6) |
| AND | 1 (4) | 2 (17)* | 0 | 3 (6) |
| Visibility | | | | |
| Stand alone | 21 (81) | 9 (75) | 3 (30) | 33 (69) |
| Integrated into resuscitation policy | 5 (19) | 3 (25) | 3 (30) | 11 (23) |
| Mentioned in other policies | 0 | 0 | 4 (40) | 4 (8) |
| Coverage | | | | |
| Local | 23 (88) | 9 (75) | 8 (80) | 40 (83) |
| Regional | 3 (12) | 3 (25) | 2 (20) | 8 (17) |

*Column does not add up to 12 because one Trust consciously and consistently referred to DNACPR/AND throughout the policy.
AND, allow natural death; AS, Ambulance service; CPR, cardiopulmonary resuscitation; DNACPR, do not attempt cardiopulmonary resuscitation; DNAR, do not attempt resuscitation.

Association, the Resuscitation Council (UK)and the RCN was the most frequently cited source of national guidance for acute and community NHS Trusts. Reference to relevant legislation (Mental Capacity Act 89%, Human Rights Act, 78%) was common. General Medical Council guidelines were highlighted less frequently (50%). Nine of 10 Ambulance Trusts additionally cited the Joint Royal College Ambulance Liaison Committee.

The national guidance that DNACPR decisions may be made on the basis of futility, overall benefit or patient refusal was incorporated into all acute and community Trust policies. Most policies clarified that the DNACPR decision related only to the act of resuscitation and did not apply to other aspects of care.

### Decision-makers and involvement of others in DNACPR decisions

All acute and community Trusts were responsible for primary DNACPR decisions while ambulance Trusts were not primary decision-makers of lasting DNACPR decisions. The review identified variation in the grade of clinical staff authorised to make an initial DNACPR decision. Authority was delegated to senior nursing staff at 7 Trusts (1 limited to MacMillan Nurses, 3 for community decisions only, 3 appropriately qualified nurses) or junior medical staff (foundation year doctors (n=3), specialist trainee doctors (n=15), most senior available/any grade (n=6) or undefined/unclear (n=3)). Accountability for DNACPR decisions rested with consultants at all acute Trusts and the most senior clinician who may be a general practitioner, consultant or nurse depending on circumstances at community Trusts. This is in line with the national guidelines which state that local policy should define who the most senior person in charge of the DNACPR decision is. However, the guidelines do not make any recommendations on the staff grade that can initiate decisions.[3] Few (8%) of the Trusts mandated medical staff to discuss decisions with others

**Table 2**  Overview of adoption of RC (UK) DNACPR forms and type of form by acute, community and AS Trusts

| | Acute Trusts (n=26), n (%) | Community Trusts (n=12), n (%) | AS Trusts (n=10), n (%) | Total, n=48, n (%) |
|---|---|---|---|---|
| DNACPR form | | | | |
| RC (UK) | 2 (7.8) | 3 (25) | 0 | 5 (10.4) |
| Modified | 4 (15.4) | 3 (25) | 1 (10) | 8 (16.7) |
| Bespoke | 16 (61.5) | 4 (33.3) | 4 (40) | 24 (50) |
| Form not sent | 4 (15.4) | 2 (16.6) | 5 (50) | 11 (22.9) |
| Type of form* | | | | |
| Paper | 24 (92.3) | 10 (83.3) | 5 (50) | 39 (81.3) |
| Electronic | 1 (3.8) | 0 | 0 | 1 (2.1) |
| Both | 1 (3.8) | 2 (16.6) | 0 | 3 (6.3) |

*Insufficient information was available in policies from 5 Trusts for this question. AS, ambulance service; DNACPR, do not attempt cardiopulmonary resuscitation; RC (UK), Resuscitation Council (UK).

within the multidisciplinary team although most (74%) recommended discussion, a fifth made no such recommendations.

Acute Trusts recommended staff to talk to patients and relatives in 100% of reviewed policies and community Trusts in 100% to patients and 92% to relatives. Guidance requiring clinical staff to assess the patient for capacity and guidance about when to consult a lasting power of attorney or independent mental capacity advocate occurred less commonly. Fifty per cent of Trusts recommended the use of a patient information leaflet. Less than 1 in 10 Trusts provided practical guidance on how to approach DNACPR decision-making in different cultures (see electronic supplement tables E2–E4 for more information on these aspects).

## Practical issues of DNACPR decision-making: validity, review and portability of DNACPR decisions

The national guidelines provide no guidance on how long DNACPR decisions should be valid and make no reference to the validity of decisions other than that the validity of decisions needs to be confirmed by receiving healthcare providers before accepting formal DNACPR decisions.[3] The duration of time over which a DNACPR decision was considered valid varied widely across local policies. DNACPR decisions from acute Trusts ranged from valid indefinitely (54%) to valid for the duration of one hospital admission only (31%) and valid until a specific point in time (8%) or up until a specified review date (4%). Similarly, in community trusts validity ranged from valid indefinitely (42%) to valid up until a specified review date (42%). Two community Trusts had no information in the policies about the validity of decisions and the duration of validity in one acute Trust was unclear. There was similarly wide variation in Trusts' approach to reviewing DNACPR decisions with timings ranging from 24 h to months (see electronic supplement for detailed information). In terms of review the guidelines state that decisions must be reviewed regularly and that the frequency of review should be determined by the health professional in charge.[3]

Ambulance services focused on the requirements to recognise a valid DNACPR decision from other organisations. The level of proof required ranged from original DNACPR form acceptable only (n=1) to photocopies with ink signature/legible signature accepted (n=2) and form not necessarily needed to be seen (n=1). Six policies did not specify the requirements on the level of proof of a valid DNACPR decision. Acceptable formats of DNACPR decisions varied considerably. Two Trusts recognised that DNACPR decisions come in a variety of formats and were willing to accept verbal and written DNACPR decisions (including letters, entry in patient notes and pro forma). Another two Trusts accepted any kind of written decisions, while one Trust stated that a decision had to be presented on a DNACPR form. Three more Trusts required specific forms only. 7/10 policies specified details that staff need to check to establish the validity of the document. These included the patient's details, the review date, a list of items if the decision is not on a pro forma, and the completeness of transport specific sections on the DNACPR form.

The portability of DNACPR decisions between organisations and healthcare settings (community/acute care) was one of the greatest sources of inconsistency and variation with limited guidance arriving from the national guidelines. These stress that 'any decisions about CPR should be communicated between healthcare professionals whenever a patient is transferred between establishments,..., or is discharged.... procedures must be in place to notify (the receiving organisation) of the patient's CPR status, and provide them with the necessary documentation.[3] (p19) DNACPR decisions were portable in 13/26 acute Trusts and 8/12 community Trusts. An additional six acute Trusts' DNACPR decisions extended to include ambulance transport. The remaining acute Trusts (n=7) but only one community Trust had a system in place through which non-portable DNACPR decisions could be communicated between providers. The level of portability of DNACPR decisions in the remaining three community Trusts was unclear. The detail with which handover systems were described in the policies was generally greater in the community Trusts than the acute Trusts. An example of a clear, unambiguous pathway, describing transfer between and within acute and community care settings of a fully portable DNACPR form is provided in box 1.

## DISCUSSION

Our review of local DNACPR policies revealed that while some isolated aspects of the national guidelines were implemented consistently into local policy, there was generally huge variation between local policies in all areas of documentation, ethical and legal issues, the decision-makers but first and foremost in the practical issues of DNACPR decision-making, that is, the validity, review and portability of decisions which greatly affect the interface between services. Some of the variation might be explained in part by the language used in the national guidelines leaving too much scope for interpretation and some might be justifiable by the need to adapt to local circumstances. Others reveal a lack of compliance with clear recommendations in the guidelines or a lack of compliance with national legislation such as the requirement to consider capacity when making DNACPR decisions. This is in contrast with the national guidelines which focus on ethical and legal issues related to DNACPR decision-making and include less guidance on the more practical issues of DNACPR decisions. The national guidelines make no reference to the validity and do not provide guidance on the frequency of review of DNACPR decisions. There is also limited guidance on the portability of decisions, that is, whether decisions should be accepted automatically by healthcare providers outside the establishment of the

**Box 1** Example of clear pathway for handover of Do Not Attempt Cardiopulmonary Resuscitation (DNACPR) decisions from one acute Trust

Review the appropriateness of the DNACPR decision before discharge

If on review the DNACPR is still considered appropriate carry decision over to the patient's care setting/home on discharge including the following considerations:

► Liaise with the patient's general practitioner and identify an agreed, appropriate community review date (ideally within 24 h), which should be documented on the DNACPR form

► Send the original DNACPR form (top white copy) with a review date in place with the patient

► Communicate and discuss sensitively with the patient

► Leave the yellow copy (second copy) in the front of the patient's medical hospital notes facilitating early consideration of resuscitation issues on any potential subsequent readmission

Communication with Ambulance Service and/or ongoing care setting must take place

Before Ambulance transfer complete appropriate section on the DNACPR form

When booking transport for DNACPR patient from acute Trust fax DNACPR form to ambulance control Contact Centre

Within community settings, hand DNACPR forms directly to the attending crew on arrival or fax to AS Contact Centre

The DNACPR status must also be recorded by the attending AS crew on the AS Patient Report Form

primary decision-maker and how the handover systems should be organised. The resulting variation in portability reflects the range of different systems that are in place across England of how DNACPR decisions are handled in English healthcare trusts.

While the aforementioned systems of handling DNACPR decisions confirm the variation in local policies of the different healthcare services there is also evidence of efforts at standardisation that are initiated by regional working groups rather than national leaders. Several regions in England have formulated unified policies with regional DNACPR forms to improve communication and handover across healthcare providers within one area. These initiatives are often sponsored by the relevant clinical commissioning groups and allow a consistent approach, easy recognition of forms and allow the decisions to cross the borders between primary and secondary healthcare. However, there is also evidence that signing up to regional policies and switching to unified forms is slow and challenging. Research is needed to understand the barriers that hinder Trusts to move from local to regional policies and forms as these barriers might also hamper the implementation of national guidelines or a possible future national policy.

The development of a national template or standardised policy to support clinicians and patients in decision-making seems intuitively attractive. This is reflected in the Tracey Case, a case between the Tracey family and the Cambridge University Hospital NHS Foundation Trust over a DNACPR decision. This case was advanced against the Secretary of State accusing him of failing to publish national guidance for clear DNACPR decision-making processes. However, in June 2014 the Court of Appeal rejected a legal necessity for a national policy.[5] A standardised policy might improve consistency in approach and reduce policy variation. It would reduce the need for doctors moving between Trusts to learn new policies each time they go to a new Trust. Furthermore, it would allow the development of some generic learning materials on DNACPR decision-making for clinicians and patients. However, the poor implementation of national guidelines and the slow and patchy recognition of regional policies might suggest that there are barriers to standardisation that need to be understood and overcome. Furthermore, there is a need to research and understand issues in the implementation and translation of local policy into local clinical practice as this would not be addressed by a standardised policy.

It is questionable that a national policy on the same lines as the national guidelines would achieve consistency in practice unless the practical issues of validity, review and portability are addressed. Local policies need more guidance on the practical aspects to standardise approaches. This would hopefully have an impact on communication and handover of DNACPR decisions between healthcare providers. However, without evidence on which system works best it would be difficult to choose one system and subsequently mandate it for all healthcare Trusts. Another approach would be to support standardisation efforts and help unified policies to spread and award regional initiatives.

While it has been suggested that incorporating DNACPR forms into an overall treatment option form (Universal Form of Treatment Options—UFTO)[6] or into treatment escalation plans[7] is associated with reduced patient harm[6] and improved communication with patients and relatives,[6 7] it is unlikely that an intervention that supports the initial decision-making process will eliminate the problems of handover and communication between healthcare providers involved in the care of the patient.

Communication of DNACPR is a major issue. The guidelines covered the appropriateness of talking to patients or relevant others in detail, however, little guidance to aid implementation into local policies was provided. Most policies described DNACPR decisions being made by clinicians in discussion with patients/relatives. Future research should explore if shared decision-making leads to better satisfaction and outcomes.

### National and international context

The current national and international literature illustrates that challenges around guidance for DNACPR decision-making is evidenced at all three levels, that is, national policy or guidelines, local policy and local practice.

While our research revealed that in England national guidelines are implemented inconsistently into local policy, a recent survey of UK DNACPR forms suggested that changes in the national guidelines were the main driver for DNACPR form amendments as it was the most frequent response.[8] However, the majority of responses included a variety of other drivers for change (55/71) which seems to confirm that the influence of the guidelines is variable across England. More concerning is that in a survey of specialist registrars over one quarter of respondents were unable to recommend a document or guideline for DNACPR decision-making.[9] The survey identified views of specialist registrars that there were no current guidelines that cover the complexities and difficulties of DNACPR decisions and that the available guidelines contradicted themselves. This also seems to support our finding that national guidelines should be improved to provide clear recommendations in all aspects of DNACPR decision-making.

Furthermore, national and international literature indicates that there are deficiencies in staff knowledge of local DNACPR policies[10] and relevant laws and regulations[11] which might be suggestive of a gap between policy and practice. For instance while our policy review suggested good compliance with national guidelines in terms of clarifying that the DNACPR decision related only to the act of resuscitation and did not apply to other aspects of care, Smith et al (2006)[10] reported that about 15% of nurses and midwifes believed that antibiotics, physiotherapy and nasogastric feeding would be inappropriate for DNACPR patients. Additional knowledge gaps included uncertainties about which professional grade can make decisions; and healthcare professionals generally did not believe that informing the multidisciplinary team of DNACPR decisions was important.[10]

In contrast, facilitators for DNACPR decision-making in the literature appear to focus on local practice. Imhof et al[12] found that interdisciplinary decisions were important to avoid conflict and non-compliance. Their research stressed that integrating nurses' views and observations into the decision would lead to more successful outcomes in compliance with resuscitation decisions. Furthermore, they recommend that decision-making should not be left to junior doctors because of the professional, human and relational expertise that is required.[12] However, involvement of nurses in the decision-making process would require additional education of nurses to fill knowledge gaps in resuscitation legislation.[11]

Further facilitators to ease DNACPR decision-making have been suggested. Myint et al[9] suggested raising patient and public awareness of resuscitation and decisions around resuscitation. This is of interest as a review of articles in the lay-press revealed that newspapers in the UK portray an overoptimistic survival rate following cardiac arrest in the out-of hospital and in-hospital setting.[13] Further claims to improve DNACPR decision-making were the implementation of a standard process

in trust policies to record patients' wishes at admission[9] while Kim do et al[14] called for a systematic standardisation of DNACPR decisions in terms of standardised forms to improve end-of-life decisions in the early stage of patients with terminal cancer. A recent systematic review confirmed these findings by ranking standardisation of DNACPR forms and structured changes to the decision-making process for instance at the time of admission as the two most promising interventions identified in the literature that could improve DNACPR decision-making.[15] Aspects for the successful implementation of a standardised approach to DNACPR decision-making were reported in the prehospital setting in two states in the USA.[16] It showed that the introduction of a standardised form together with a core protocol at the same time as in-service training proved valuable for its success.[16] Approach in the Emergency Medical Services boards to gain approval also helped protect from the development of multiple forms as well as using unambiguous language and ensuring immunity from litigation. Finally, education of the public through news articles about the service was thought to be a particular strength of the programme.

In terms of policy and guidance published research is less specific in their recommendations. A survey of 298 Irish consultant physicians found that only 21% were aware of a formal resuscitation policy in their hospital which is why the authors called for a national policy for resuscitation decision-making to facilitate more widespread formulation of local policies.[17]

Overall, the national and international literature tends to address issues around the initial decision-making process including the involvement of nurses, standardisation and lack of awareness of legislation. However, there appears to be a lack of research addressing issues relating to the interface of healthcare settings for safe handover of DNACPR decisions.

## Strengths and limitations

While this research reviewed only a sample of local policies, these covered all areas of England and met our objectives as this sample sufficiently identified huge variation and a worrying number of different systems of how Trusts deal with DNACPR decisions across England. Furthermore, the review included policies from different types of Trusts which was important as these concern healthcare providers with different responsibilities. The ambulance services for instance are not primary decision-makers of lasting DNACPR decisions and are therefore more concerned with the handover and checking the validity of decisions. This relates to lasting DNACPR decisions only. AS staff attending patients in cardiac arrest still have to make immediate but informed decisions on whether to start CPR or when to stop CPR. However, this research concentrated on lasting DNACPR decisions that AS face when accepting a DNACPR patient from acute Trusts or the community.

Community Trusts are an assemblage of diverse healthcare providers. Their concern, therefore, concentrates on

the handover, communication and consistent documentation of decisions within the Trust. This might explain why community services tended to describe handover systems in more detail than acute Trusts and why the proportion of portable decisions and unified policies was greater among community than acute Trusts.

One potential limitation of our research is that only about 30% of the extracted data were checked by a second reviewer. This would be of concern if it applied to the subjective questions covering the questions around the validity and portability of DNACPR decisions. However, following good agreement between the two reviewers on early data extractions, the decision was made to limit the second review to the more subjective questions but include all policies in this focused second review.

As our research only addressed the implementation of national guidelines into local policies it does not highlight shortfalls in the implementation of national and local guidance into local practice. There is a need to investigate issues and possible solutions to improve uptake of guidance into local practice.

## CONCLUSIONS

This review identified significant variation in English NHS Trusts approaches to DNACPR decision-making. Gaps identified included practical guidance around when and how to communicate DNACPR decisions and in relation to the portability/transferability of decisions between healthcare settings. There is a need for greater consistency in Trusts approaches to DNACPR decision-making.

**Contributors** GDP conceived the study and obtained funding. KF developed the detailed protocol and led the primary data extraction with help from RAF. KF drafted the paper with GDP and RAF revising it critically for important intellectual content. All authors approved the final version.

**Funding** This project was funded by the National Institute for Health Research (Health Service Research and Delivery Programme, project number 12/5001/55). The views and opinions expressed therein are those of the authors and do not necessarily reflect those of the (Health Service Research and Delivery Programme), NIHR, NHS or the Department of Health. GDP is a Director of Research for the Intensive Care Foundation.

**Competing interests** GDP is a volunteer member of the Resuscitation Council (UK). The Resuscitation Council UK coauthored the joint national statement on DNACPR decisions.

**Provenance and peer review** Not commissioned; externally peer reviewed.

**Data sharing statement** No additional data are available.

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
