## [Reviewer comments · BMJ Open]

Some articles will have been accepted based in part or entirely on reviews undertaken for other BMJ Group journals. These will be reproduced where possible.

ARTICLE DETAILS

TITLE (PROVISIONAL)	Variation in local Trust Do Not Attempt Cardiopulmonary Resuscitation policies: a review of 48 English health care trusts
AUTHORS	Perkins, Gavin; Freeman, Karoline; Field, Richard

VERSION 1 - REVIEW

REVIEWER	Claud Regnard St. Oswald's Hospice, Newcastle-upon-Tyne, NE3 1EE, UK
REVIEW RETURNED	06-Sep-2014

GENERAL COMMENTS	This is an important paper that exposes a remarkable variation in the practice of making and documenting CPR decisions, with implications for national guidance. 1) The background should include an observation on the segregation of CPR from all other acute, life-saving treatments in terms of decision-making.2) The Mental Capacity Act is a legal requirement in England and Wales. The important finding that 42% of acute trusts and 25% of community trusts have no policy requirement to consider capacity when making CPR decisions deserves stronger emphasis in the results and more space in the discussion.3) The recent Tracey Court of Appeal judgement should be mentioned in the discussion, in particular its rejection of a national policy.4) One of the compliance criteria used was futility, but it is unclear how this was interpreted by the researchers.5) The authors have avoided recommending a national policy, but these results suggest regional policies as a minimum with a national approach as a reasonable goal.6) Research recommendations should include exploring the place of CPR decisions in shared decision making. To avoid delaying publication, the suggested revisions can be made with minimal changes to the text.
---

REVIEWER	H.R.W. Pasman VU university medical center, Amsterdam, The Netherlands
REVIEW RETURNED	17-Sep-2014

GENERAL COMMENTS	Re 9. and 10. I think this paper can be approved by incorporating the supplement data into the main text and that for every topic it is also described what the national guidelines state about this topic (such as on definition, clinical staff authorisation and review of a
---

	document). Now it is described in the discussion what is (not) in the national guidelines, but i would prefer to know more about the content of the national guidelines and the (dis)agreement with local ones. Methods: I would like to know more about the sample size: How many acute hospital Trusts and community health Trusts does the UK have in total and % of included in this study (26/xxx acute hospital Trusts, and 12/xxx community health Trusts)
--	--

REVIEWER	Bernard Foex Emergency Department Manchester Royal Infirmary Manchester UK
REVIEW RETURNED	22-Sep-2014

GENERAL COMMENTS	In the Introduction you mention the Human Rights Act in relation to the right to be free from inhuman and degrading treatment. Are you suggesting that CPR is either of these? Lady Butler Sloss reminded us some years ago that inhuman and/or degrading treatment would have to be perceived that way by a patient. Those receiving CPR are unlikely to experience their treatment as either inhuman or degrading. We might consider it degrading but that falls outside the Human Rights Act. While the Human Rights Act covers the right to life it does not guarantee a right to be kept alive. The relevance of the HRA to DNACPR decisions needs some clarification. Towards the end of page 3 you write, “the guidelines provide general principles that require tailoring to local circumstances, which suggests that there may be room for interpretation of national guidelines when implemented into local policy.” So you would expect variation, particularly between different types of trusts. I don’t think you addressed possible local circumstances as a source of variation in the discussion. If electronic supplementary information is going to be provided might it be worth adding in the data extraction form and explaining how the quantitative data and qualitative data were synthesized. On page 5 you comment that none of the forms required the patient to give consent. This needs some explanation or qualification. As it stands it reads like a reproach. I don’t think there has ever been a suggestion that patients should sign such forms, in which case there is no reason why it should feature. If the authors have an opinion then this comment and the opinion should be included in the Discussion. Page 7 You mention DNACPR decision making in different cultures (see electronic supplement). There didn’t seem to be any detailed in the supplementary information supplied. On page 9 The last sentence of the 2nd paragraph is not clear, especially, “understand the barriers that hinder Trusts to go unified as these might be equivalent to barriers to the implementation of
--

	national guidelines or a possible future national policy.”
	Typos
	Page 4, First line “ we obtained acute care Trust lists” ??
	Page 6, Last sentence, “Few (8%) of the Trusts ...”
	Page 9, 2nd paragraph, “of efforts at standardization ...”
	Page 9, 3rd paragraph, “Such a measure might ...”

VERSION 1 – AUTHOR RESPONSE

Reviewer 1

This is an important paper that exposes a remarkable variation in the practice of making and documenting CPR decisions, with implications for national guidance.

Thank you.

1) The background should include an observation on the segregation of CPR from all other acute, life-saving treatments in terms of decision-making.

We have included this observation. (see page 4)

2) The Mental Capacity Act is a legal requirement in England and Wales. The important finding that 42% of acute trusts and 25% of community trusts have no policy requirement to consider capacity when making CPR decisions deserves stronger emphasis in the results and more space in the discussion. We have given this due consideration and have highlighted this issue in the discussion. However, the results section lists the outcomes of all questions without any weight, which we feel is appropriate.(see page 9)

3) The recent Tracey Court of Appeal judgement should be mentioned in the discussion, in particular its rejection of a national policy. We have included the Tracey Case in the Discussion. (see page 10)

4) One of the compliance criteria used was futility, but it is unclear how this was interpreted by the researchers.

This was included as a yes / no question alongside Benefit / Burden and Refusal of CPR. We reviewed the policies to identify whether policies mention the three situations in which DNACPR decisions are appropriate but did not investigate whether their definition matches the definition in the national guidelines.

We adopted the interpretation of the national guidelines.

We have amended the text to reflect this. (see page 7)

5) The authors have avoided recommending a national policy, but these results suggest regional policies as a minimum with a national approach as a reasonable goal.

This is correct. We go on to highlight the need for future research to identify the barriers to

what intuitively sounds like the sensible solution to the diversity shown in this study. (see page 10)

6) Research recommendations should include exploring the place of CPR decisions in shared decision making.

We have added the following statement "Future research should explore if shared decision-making leads to better satisfaction and outcomes." (see page 11)

Reviewer 2

1) I think this paper can be approved by incorporating the supplement data into the main text

We followed the authors' guidance of the BMJ Open which states: "We recommend your article does not exceed 4000 words, with up to five figures and tables. This is flexible, but exceeding this will impact upon the paper's 'readability'." We feel that by using the option to have material in the supplement, the paper is more accessible, it includes all main results with additional information available as suppl. information for the interested reader.

2) and that for every topic it is also described what the national guidelines state about this topic (such as on definition, clinical staff authorisation and review of a document). Now it is described in the discussion what is (not) in the national guidelines, but i would prefer to know more about the content of the national guidelines and the (dis)agreement with local ones.

We have included statements from the Joint statement for the main topics in the main text. see pages 6, 7, 8, and 9

3) Methods: I would like to know more about the sample size: How many acute hospital Trusts and community health Trusts does the UK have in total and % of included in this study (26/xxx acute hospital Trusts, and 12/xxx community health Trusts)

We have added denominators in the main text. p5

4) In the Introduction you mention the Human Rights Act in relation to the right to be free from inhuman and degrading treatment. Are you suggesting that CPR is either of these? Lady Butler Sloss reminded us some years ago that inhuman and/or degrading treatment would have to be perceived that way by a patient. Those receiving CPR are unlikely to experience their treatment as either inhuman or degrading. We might consider it degrading but that falls outside the Human Rights Act. While the Human Rights Act covers the right to life it does not guarantee a right to be kept alive. The relevance of the HRA to DNACPR decisions needs some clarification.

We believe that CPR treatment can be perceived that way by a patient who is at the end of his / her life and wishes to die in the place of his / her choice and refuses CPR treatment for that very reason. The Joint statement heavily references the Human Rights Act and 78% of reviewed policies thought that the Human Rights Act is relevant for CPR decisions.

We have added a further fundamental right in the introduction that is relevant to DNACPR decision making.

The Joint statement: Policies and individual decisions about CPR must comply with the Human Rights Act 1998. This Act incorporates the bulk of the rights set out in the European Convention on Human Rights into UK law.

In order to meet their obligations under the Act, health professionals must be able to show that their decisions are compatible with the human rights set out in the Articles of the Convention. Provisions particularly relevant to decisions about attempting CPR include the right to life (Article 2), to be free from inhuman or degrading treatment (Article 3), to respect for privacy and family life (Article 8), to freedom of expression, which includes the right to hold opinions and to receive information (Article 10) and to be free from discriminatory practice in respect of these rights (Article 14). The spirit of the Act, which aims to promote human dignity and transparent decision making, is reflected in these ethical guidelines. 3

5) Towards the end of page 3 you write, “the guidelines provide general principles that require tailoring to local circumstances, which suggests that there may be room for interpretation of national guidelines when implemented into local policy.” So you would expect variation, particularly between different types of trusts. I don’t think you addressed possible local circumstances as a source of variation in the discussion.

We have considered this in the discussion and amended the document accordingly. p9

6) If electronic supplementary information is going to be provided might it be worth adding in the data extraction form and explaining how the quantitative data and qualitative data were synthesized.

We have added the data extraction table to the electronic supplement.

See reply editor’s comments for additional information on the synthesis.

7) On page 5 you comment that none of the forms required the patient to give consent. This needs some explanation or qualification. As it stands it reads like a reproach. I don’t think there has ever been a suggestion that patients should sign such forms, in which case there is no reason why it should feature. If the authors have an opinion then this comment and the opinion should be included in the Discussion.

We have removed this statement as we agree it is distracting to the main message from the paper.

It is interesting to note (but outside the scope of this paper) that during the development of the RC(UK) national DNACPR form that the legal advisors strongly advocated the form should be signed.

8) Page 7 You mention DNACPR decision making in different cultures (see electronic supplement). There didn’t seem to be any detailed in the supplementary information supplied.

The guidance on decision making for different cultures was included as a yes / no question and is covered in Table E1.

The main text has been amended to avoid confusion what the additional information relates to. 7

9) On page 9 The last sentence of the 2nd paragraph is not clear, especially, “understand the barriers that hinder Trusts to go unified as these might be equivalent to barriers to the implementation of national guidelines or a possible future national policy.”

We have rephrased the sentence. see page 10

10) Typographical errors

These have been corrected.